# Triple-Negative Breast Cancer and Emerging Therapeutic Strategies: ATR and CHK1/2 as Promising Targets

**DOI:** 10.3390/cancers16061139

**Published:** 2024-03-13

**Authors:** Amalia Sofianidi, Ecaterina E. Dumbrava, Konstantinos N. Syrigos, Azadeh Nasrazadani

**Affiliations:** 1Oncology Unit, Third Department of Internal Medicine, Sotiria General Hospital for Chest Diseases, National and Kapodistrian University of Athens, 11527 Athens, Greece; amsof.00@gmail.com (A.S.); ksyrigos@med.uoa.gr (K.N.S.); 2Department of Investigational Cancer Therapeutics, The University of Texas MD Anderson Cancer Center, Houston, TX 77030, USA; 3Department of Breast Medical Oncology, The University of Texas MD Anderson Cancer Center, Houston, TX 77030, USA

**Keywords:** triple-negative breast cancer (TNBC), checkpoint kinase 1 (CHK1), checkpoint kinase 2 (CHK2), ataxia telangiectasia and Rad3-related (ATR), DNA damage response, targeted therapy

## Abstract

**Simple Summary:**

Triple-negative breast cancer (TNBC) is recognized for its heightened aggressiveness compared to other breast cancer subtypes. Given the lack of an associated biomarker or molecular target, therapeutic options are limited. In an effort to expand the therapeutic landscape of TNBC, interest has been mounting in the exploitation of the DNA damage response pathway (DDR), particularly with regard to agents that inhibit ATR kinase and Checkpoint Kinases 1/2 (CHK1/2). ATR and CHK1/2 inhibitors show potential as prospective treatment options for TNBC by interfering with the cell cycle regulation of cancer cells. Initial findings indicate that the co-administration of ATR and CHK1/2 inhibitors alongside chemotherapy effectively inhibits tumor growth in TNBC. In this literature review, we explore ATR and CHK1/2 inhibition as a promising therapeutic approach in the management of TNBC and highlight challenges arising during the administration of these novel agents.

**Abstract:**

Worldwide, breast cancer is the most frequently diagnosed malignancy in women, with triple-negative breast cancer (TNBC) being the most aggressive molecular subtype. Due to the dearth of effective therapeutic options for TNBC, novel agents targeting key mechanisms and pathways in cancer cells are continuously explored; these include ATR inhibitors, which target the ATR kinase involved in the DNA damage response (DDR) pathway, and CHK1/2 inhibitors, which target the Checkpoint Kinase 1/2 (CHK1/2) involved in cell cycle arrest and DNA repair. ATR and CHK1/2 inhibitors show potential as prospective treatments for TNBC by focusing on the DDR and interfering with cell cycle regulation in cancer cells. Preliminary preclinical and clinical findings suggest that when combined with chemotherapy, ATR and CHK1/2 inhibitors demonstrate significant anti-proliferative efficacy against TNBC. In this article, we introduce ATR and CHK1/2 inhibitors as promising therapeutic approaches for the management of TNBC. Preclinical and clinical studies performed evaluating ATR and CHK1/2 inhibitors for the treatment of TNBC and associated challenges encountered in this context to date are reviewed.

## 1. Introduction

On a worldwide scale, breast cancer (BC) accounts for approximately 30% of all cancers occurring in females [1,2] and is the most frequently diagnosed malignancy [2,3,4,5], accounting for over two million cases each year [6,7]. After lung cancer, BC ranks as the second leading cause of cancer-related deaths among women overall, but among Black and Hispanic women, it holds the unfortunate distinction of being the leading cause of cancer death [8,9]. With regard to molecular subtypes, BC can be broadly classified into three groups: hormone receptor-positive (HR+) (expressing estrogen receptor (ER+) or progesterone receptor (PR+)), human epidermal growth factor receptor 2-positive (HER2+), or triple-negative breast cancer (TNBC) [10,11]. The HER2 low entity has recently become of increasing interest and is characterized by low levels of the HER2 protein [12]. TNBC cells lack the expression of ER, PR, and HER2 [13,14]. It is a clinically and biologically heterogeneous [15,16,17,18] disease associated with a poor prognosis and aggressive behavior [13], reflecting an area of unmet need in which the development of effective therapies is of outmost importance for patients with this entity. The diagnosis of TNBC is based on immunohistochemistry (IHC) [19] and fluorescence in situ hybridization (FISH) technique when applicable [20].

The standard chemotherapy regimen for TNBC usually consists of anthracyclines and taxanes [21]. In the case of women diagnosed with BRCA-mutated TNBC and whose cancer has become resistant to standard chemotherapy, alternative platinum-based chemotherapy agents (such as cisplatin or carboplatin) or targeted medications known as PARP inhibitors (olaparib or talazoparib) may be deliberated as treatment options [22]. When cancer cells express the PD-L1 protein, initial treatment may involve a combination of immunotherapy (pembrolizumab) and chemotherapy [23]. For advanced TNBC in which at least two other drug treatments were already tried, the antibody–drug conjugate sacituzumab govitecan might be an option [24]. Despite the utilization of adjuvant radiation, targeted therapies and, more recently, immunotherapy recurrence rates remain high [25]. A range of promising agents and combinations are currently being developed [26]; notable among them are ATR inhibitors, which target the ATR kinase involved in the DNA damage response pathway (DDR) [27,28], and CHK1/2 inhibitors, targeting Checkpoint Kinase 1/2 (CHK1/2) involved in cell cycle arrest and DNA repair [28,29]. In this review, we introduce ATR and CHK1/2 inhibitors as promising therapeutic options for the management of TNBC. Preclinical and clinical studies performed to date evaluating ATR and CHK1/2 inhibitors and associated challenges encountered in this context are reviewed. 

## 2. DNA Damage Response (DDR) and TNBC

Millions of DNA-damaging lesions occur daily on a cellular level due to various stressors [30]. The DDR pathway is a complex network of cellular processes that safeguard the integrity of DNA and ensure proper repair. When DNA damage occurs, DDR is activated to detect damage, initiate repair mechanisms, and, if necessary, induce cell cycle arrest or cell death to prevent the propagation of potentially harmful mutations, thus ensuring the maintenance of genomic stability and survival [31,32,33]. TNBC is characterized by the activation of oncogenes, such as *MYC* and cyclin E amplification, which in turn creates replication stress in the cells, leading to an accumulation of DNA damage and reliance on the DDR for survival [30]. In the context of TNBC, the DDR is often dysregulated, making the cancer cells more susceptible to DNA damage and potentially more reliant on specific DNA repair mechanisms. This characteristic is exploited in targeted therapies in development for TNBC.

Currently, the most-studied DDR pathways in TNBC are the ATR-CHK1-WEE1 pathway and homologous recombination repair (HRR) [34]. ATM (ataxia-telangiectasia-mutated) and ATR (ataxia-telangiectasia and Rad3-related) are activated in response to DNA damage, acting as sensors and initiating the DDR [35]. When ATR is activated in cells during S-phase, it hinders the initiation of replication origins and limits DNA synthesis, while its activation in G2 cells encourages the arrest of the cell cycle at the G2/M transition [35,36]. ATR recognizes the single-stranded DNAs (ssDNAs) that arise from replication stress and DNA damage related to radiation or chemotherapy and then phosphorylates CHK1 on Ser345 and Ser317. CHK1 phosphorylates CDC25a, triggering the degradation of CDC25a in a ubiquitin/proteasome-dependent manner [37,38] and serving as an intra-S phase checkpoint [38,39]. An active cyclin B/CDK1 complex is essential to initiate the process of mitosis [40]. Activated CHK1 decreases or inactivates CDC25c phosphatase in the nucleus [41,42], preventing the activation of CDK1 in the nucleus and serving as a G2/M phase checkpoint. WEE1 is simultaneously activated, which in turn inactivates CDK1 through phosphorylating its inhibitory phosphorylation site at Tyr15. Thus, the activation of the ATR-CHK1 pathway leads to the inhibition of cyclin B/CDK1 complex through the activation of WEE1 and suppression of CDC25c. This leads to the activation of the G2/M checkpoint, effectively halting cells with DNA damage from entering mitosis [43,44]. 

ATR is also involved in HRR [45,46], a pathway that repairs DNA double-stranded breaks (DSBs) [47]. Paradoxically, although ATR is known to inhibit CDKs, CDK activity is necessary for efficient HRR. CDKs facilitate the trimming of DNA ends at DSBs, which leads to the formation of the required ssDNA crucial for HRR [48,49]. ATR promotes HRR by phosphorylating BRCA1 and CHK1 and activating another two key proteins, RAD51 and BRCA2 [50,51]. During HRR, BRCA1 brings the PALB2-BRCA2 complex to the sites of DSBs [52,53,54], and this process leads to the formation of the RAD51 filament [55,56,57,58]. Rad51 functions as a DNA-binding protein that has the capacity to oversee the activities of nucleases, helicases, DNA translocases, and signaling proteins. This enables Rad51 to play a regulatory role in situations of replication stress, including tasks such as facilitating fork reversal and restoring repaired forks [59]. The phosphorylation of H2AX holds a crucial position within DDR, being essential for the gathering of DNA repair proteins at locations with compromised chromatin. This process also triggers the activation of checkpoint proteins responsible for halting the progression of the cell cycle [60,61,62]. Figure 1 provides a schematic of key proteins in DDR and targets of inhibitors in development, which are detailed in the following section.

Frequently, TNBC cells exhibit impaired p53 signaling and/or loss of the RB protein, leading to an ineffective G1/S checkpoint [63,64,65]. TNBC cells are, thus, more reliant on the G2/M checkpoint, particularly on ATR-CHK1-dependent DDR, for survival [66]. Interestingly, p53 deficient tumors demonstrate sensitivity to blockade by ATR or CHK1 inhibitors [67], positioning these agents as potentially attractive agents for the management of TNBC. 

BRCAness is a phenotype resembling mutations of germline BRCA1/2 DNA repair genes, which results in HRR deficiency [68]. Recent preclinical studies showed that certain oncogenes, such as *RAS* and *PI3K*, and the androgen receptor may directly or indirectly influence HRR activity. Regardless of DDR gene mutation status, it was preclinically demonstrated among advanced-stage or high-risk prostate cancer cell lines that simultaneous or sequential targeting of the androgen receptor with PARP and/or ATR inhibition results in synthetic lethality [69,70,71]. Beyond DDR alterations that result in a BRCAness phenotype, TNBC is also associated with a “chemical BRCAness” phenotype resulting from such oncogene mutations [72]. As anticipated, TNBC cells lack a sufficient response to PARP and/or ATR inhibition monotherapy in the absence of DDR alterations. 

## 3. Targeting DDR in TNBC

### 3.1. ATR Inhibitors in TNBC

ATR inhibitors showed promise as a targeted therapy approach for the management of TNBC with multiple agents currently at various stages of development. Elevated ATR protein levels in breast tumors are more likely to be seen in patients with advanced tumor stage and increased occurrence of lymphovascular invasion [50]. There are multiple ATR inhibitors (ATRis) undergoing clinical studies, including VX-970 (also known as VE-822, M6620, or berzosertib), AZD6738 (ceralasertib), VX-803 (M4344 or gartisertib), BAY1895344 (elimusertib), and the promising ATRi RP-3500 (camonsertib). Other ATRis that are currently in phase I clinical trials regarding solid tumors are M1774 from EMD Serano and ART0380 from Artios. The structure of berzosertib is based on that of VE-821 but with improved potency and absorption, distribution, metabolism, and excretion (ADME) properties that allowed for use in in vivo studies [73,74]. VE-821 lacked the “drug-like” characteristics of increased potency and selectivity against ATR necessary for proceeding into clinical trials [75,76]; however, it still serves as an appropriate template for designing new ATRis and as a good standard for in vitro biological activity assays and cell cultures [77]. As described by Jo et al., the four ATR inhibitors in clinical development can be ranked by potency as Camonsertib (RP-3500) > M4344 ~ BAY1895344 > berzosertib (M6620/VX-970) > ceralasertib (AZD6738) [78,79].

#### 3.1.1. Berzosertib

The pioneering ATR inhibitor to undergo clinical trials was VX-970 (also known as VE-822, M6620, or berzosertib), which is intravenously administered [80]. Tu et al. demonstrated berzosertib to be a tumor-specific radiosensitizer for TNBC. Patient-derived xenografts (PDXs) established from patients with chemoresistant TNBC additionally proved to be highly radiosensitized [81]. There is an ongoing Ib clinical trial testing the addition of berzosertib to radiation therapy for chemotherapy-resistant triple-negative and estrogen and/or progesterone receptor-positive, HER2-negative BC (NCT04052555), and the first results were expected by the end of 2023. The rationale behind this combination lies in berzosertib’s potential to halt tumor cells via the inhibition of key enzymes crucial for cell growth. Combined with radiation therapy, the efficacy of cell death mediated by berzosertib is anticipated to be enhanced. This combined approach may also lead to prolonged tumor stabilization compared to the use of radiation therapy alone.

The first-in-human, open-label, phase I trial with berzosertib is completed (NCT02157792), aiming to evaluate the safety, tolerability, pharmacokinetics (PK), and preliminary antitumor activity of berzosertib in combination with gemcitabine, with or without cisplatin in solid tumors. In the cohort utilizing berzosertib with cisplatin, partial response (PR) was seen in one out of four TNBC patients, but there was progression after four treatment cycles [82]. In the cohort utilizing berzosertib with gemcitabine, PR was seen in one patient with HR+, HER2-, germline BRCA2 mutation-positive metastatic BC (mBC) [83]. In the phase I dose expansion cohort, berzosertib with cisplatin in metastatic TNBC (mTNBC) was tested among 47 patients with advanced mTNBC. The objective response rate (ORR) [90% confidence interval (CI)] was 23.4% of patients (13.7, 35.8). The best overall response (BOR) was complete response (CR) for two (4.3%) patients; one patient had a TP53 mutation, and the other had unknown mutational status, PR for nine (19.1%) patients, and stable disease (SD) for 18 (38.3%) patients. The median duration of response (DOR; [90% CI]) was 6.0 months (5.1, not defined). Median progression-free survival (PFS; [90% CI]) was 4.0 months (2.8, 6.0), and median overall survival (OS; [90% CI]) was 12.4 months (7.8, 14.5). Grade ≥ 3-related treatment-emergent adverse events occurred in more than 10% of patients: neutropenia (*n* = 18), anemia (*n* = 12), vomiting (*n* = 6), thrombocytopenia (*n* = 6), and febrile neutropenia (*n* = 2) [84]. In the cohort combining berzosertib with carboplatin, one patient was enrolled in the berzosertib monotherapy arm and one patient in the berzosertib–carboplatin arm. The patient in the combination arm had mBC and experienced dose-limiting toxicity when berzosertib 90 mg/m^2^ once weekly along with carboplatin were administered. This patient, who underwent extensive pre-treatment with 11 lines of cytotoxic therapies, experienced febrile neutropenia but subsequently recovered without any lasting effects [85].

#### 3.1.2. Ceralasertib

AZD6738 (ceralasertib), a selective small-molecule ATR inhibitor that is orally available [80], was recently introduced as the second ATR inhibitor in clinical trials. Jin et al. demonstrated that inhibition of ATR by ceralasertib in TNBC cell lines can enhance the growth suppression of TNBC by AZD1775, which is a selective WEE1 inhibitor. This response is attributed to suppressed DNA damage repair and excessive replication stress, thereby causing increased DNA damage measured by the accumulation of the DNA DSB marker γH2AX [86]. Ceralasertib showed synergistic efficacy when combined with agents known to induce replication fork stalling and collapse, such as carboplatin and irinotecan, as well as the PARPi olaparib. In a BRCA2-mutant patient-derived TNBC xenograft model, complete tumor regression was achieved with three to five days of daily ceralasertib per week, concurrent with olaparib [87]. 

In a phase I modular study of ceralasertib in combination with carboplatin, olaparib, or durvalumab in patients with advanced cancers (NCT02264678), preliminary data showed that two BRCA mutant TNBC patients achieved RECIST PRs [88]. The most common treatment-related adverse effects related to ceralasertib administration were anemia, thrombocytopenia, and neutropenia. Dose-limiting toxicities were thrombocytopenia and neutropenia [88]. VIOLETTE (NCT03330847) is a randomized phase II study aimed to assess the DDR inhibitors ceralasertib or AZD1775 in combination with olaparib versus olaparib monotherapy in patients with mTNBC. A total of 226 patients with mTNBC were randomized to be treated in the second or third line between the combination of olaparib and ceralasertib versus olaparib alone, but no significant difference in ORR or PFS with the combination was observed [89]. The lack of benefit may be attributed to the fact that the trial included PARPi naïve patients [52]. Another phase II trial (NCT04090567) evaluated the efficacy of the combination of olaparib with cediranib or ceralasertib in patients with germline BRCA-mutated advanced or mBC. Given the higher propensity for BRCA1/2 mutations in ER-/PR-BC [90,91,92], ceralasertib may prove to be efficacious in the TNBC setting. PHOENIX is a phase IIa study utilizing ceralasertib (NCT03740893) to assess whether short exposure to a DDR inhibitor and/or anti-PD-L1 immunotherapy in a preoperative window of opportunity in patients with post-neoadjuvant chemotherapy high residual disease generates a signal of antitumor biological activity within residual disease. Another current phase II study (NCT03801369) is evaluating the novel combination of olaparib with durvalumab, selumetinib, or capivasertib or ceralasertib alone in the mTNBC setting. 

#### 3.1.3. Gartisertib

Gartisertib (VX-803 or M4344) is an ATP-competitive, orally active, and selective ATR inhibitor that potently inhibits ATR-driven phosphorylated CHK1 phosphorylation [75,93]. In studies evaluating its efficacy as a monotherapy treatment, gartisertib demonstrates the ability to halt tumor growth and induce regression in models characterized by alternative lengthening of telomeres (ALT). When used alongside PARPi, a decrease in tumor size was observed in xenograft models of TNBC [93,94]. The first-in-human study (NCT02278250) of gartisertib explores the safety, tolerability, pharmacokinetics, and pharmacodynamics of gartisertib in monotherapy as well as in combination with chemotherapy in solid tumors. Gartisertib was generally well-tolerated at lower doses; however, unexpected liver toxicity prevented further dose expansion, potentially limiting antitumor activity. Elevations in blood bilirubin and liver enzymes were not reported with other ATR inhibitors in clinical development, including berzosertib and M1774. Based on the results of the present study, the development of gartisertib was halted in favor of advancing the orally administered ATR inhibitor M1774, which demonstrated higher exposure with fewer associated toxic effects [95].

#### 3.1.4. Elimusertib

Elimusertib (BAY-1895344) is a potent, orally active, and selective ATR inhibitor [96,97], which demonstrates potent anti-proliferation efficacy as monotherapy in a variety of xenograft models of ovarian and colorectal cancer. It also causes complete tumor remission in mantle cell lymphoma models [97], but it has not been studied extensively in TNBC. A phase I clinical trial (NCT04491942) is currently ongoing to identify the optimal dose and side effects of elimusertib in combination with chemotherapy in patients with solid tumors, enrolling patients with TNBC as well. Four patients with BC were enrolled with preliminary data demonstrating that the administration of elimusertib via oral treatment is well tolerated with antitumor efficacy in patients with advanced solid tumors, especially those with ATM deleterious mutations and/or a loss of ATM protein. Additionally, it exhibits activity in BRCA1-mutant cancers that developed resistance to PARPi, particularly in individuals who underwent extensive prior treatments [98]. To date, the most common adverse events related to elimusertib administration are manageable and reversible hematologic toxicities [98].

#### 3.1.5. Camonsertib

Camonsertib (RP-3500) is a potential best-in-class oral small-molecule inhibitor of ATR. Its antitumor efficacy in DDR-deficient advanced solid tumors is currently explored in the phase I TRESR trial (NCT04497116). Preliminarily, 17 patients with BC were enrolled. One patient with polyclonal BRCA1 somatic-mutated TNBC was previously treated with two prior regimens of olaparib (monotherapy and in combination with the WEE1 inhibitor adavosertib). Upon camonsertib treatment, all BRCA1 mutations measured through ctDNA levels declined in blood but then rebounded before the progression of non-target lesions at 29 weeks of treatment. Overall, camonsertib was well tolerated in the study; anemia was the most common drug-related toxicity (32% Grade 3) [79]. There is an ongoing phase I/II clinical trial (NCT04972110) aiming to evaluate the safety and tolerability of niraparib or olaparib in combination with RP-3500 (camonsertib), in patients with eligible advanced solid tumors.

### 3.2. CHK1 Inhibitors in TNBC

DNA damage and replication stress activate ATR, which in turn phosphorylates multiple substrates, including the kinase CHK1, to regulate cell-cycle progression and replication fork stability and initiate HRR [41]. The overexpression of CHK1 was reported in various human cancers, such as colon, breast, stomach, cervical, and liver cancer [99,100,101,102,103]. Gene expression analysis revealed that the *CHK1* gene is significantly overexpressed in TNBC as compared to non-TNBC and benign lesions [104], while Verlinden et al. demonstrated that CHK1 expression is significantly higher in grade 3 breast carcinomas showing a triple negative ER−/PR−/HER-2− phenotype compared with other grade 3 tumors [100]. Furthermore, Kim et al. demonstrated that the metastatic potential of cancer cells was increased by CHK1 through epithelial-to-mesenchymal transition (EMT) marker levels, enhanced migration and invasion activity [105]. CHK1 inactivation by knockdown reduces tumor growth [106,107] and chemotherapy resistance in TNBC human cell lines [108,109]. Using siRNAs to knockdown CHK1 showed significant inhibition of TNBC cell growth when compared to other TNBC-specific genes, such as ribonucleotide reductase 1 and 2 (*RRM1* and *2*) genes, without affecting normal human breast epithelial cells [104,110]. Altogether, these data highlight CHK1 inhibitors as a promising therapeutic option for patients with TNBC. CHK1/2 inhibitors target both CHK1 and CHK2; meanwhile, there are also selective CHK1 and CHK2 inhibitors. There are six CHK1 inhibitors (CHK1is) currently undergoing clinical studies, including GDC-0425, AZD7762, V158411, Prexasertib, UCN-01, and MK-8776.

#### 3.2.1. GDC-0425

GDC-0425 (RG-7602) is an orally bioavailable, highly selective small molecule inhibitor of CHK1 [111,112]. In preclinical studies, the combination of GDC-0425 and gemcitabine was found to interfere with S and G2 checkpoints, inducing premature entry into mitosis and leading to mitotic catastrophe [113,114]. A phase I study (NCT01359696) of the CHK1 inhibitor GDC-0425 in combination with gemcitabine in patients with refractory solid tumors showed two confirmed PRs in two out of five patients with TP53-mutated TNBC [112].

#### 3.2.2. AZD7762

AZD7762 potently inhibits CHK1 and CHK2 [115], but its main antitumor effect is caused by CHK1 inhibition [110,115]. Min et al. showed therapeutic synergism with gemcitabine and AZD7762 combination treatment in a TNBC cell line. The addition of birinapant (TL32711) improved the response to gemcitabine/AZD7762 combination therapy in TNBC cell lines [116]. In vitro and xenograft studies revealed the sensitivity of TNBC to AZD7762 was enhanced by RB loss [117]. 

Neoadjuvant chemotherapy (NACT) is a strategy that involves administering chemotherapy to reduce the size of a tumor prior to definitive surgical management [118] and is frequently utilized in the management of TNBC [119]. Standard-of-care chemotherapy for patients with TNBC includes both an anthracycline and a taxane [120]. Notably, however, Zhu et al. demonstrated the combination treatment of carboplatin with CHK1 inhibitor AZD7762 synergistically inhibits TNBC cell growth in multiple TNBC cell lines in vitro. Prolonged treatment with carboplatin-induced cell mitotic arrest and cells failed to initiate the G2-M transition [121]. However, the development of AZD7762 was halted in 2014 owing to unpredictable cardiac toxicity [122].

#### 3.2.3. V158411

V158411 is a potent, selective inhibitor of recombinant CHK1 and CHK2 kinases [123] proven to be most effective against TNBC cancer cell lines as compared to other breast cancer types. Bryant et al. discovered that the ovarian cancer cell line under scrutiny had high expression levels of pChk1 (S296), and it was relatively resistant to growth inhibition by CHK1 inhibitors. Further work is needed to understand the relative resistance of this cell line to CHK1 inhibition and its potential linkage with TNBC cell lines. In the careful selection of patients who would derive the most significant benefits from CHK1 inhibitor therapy, the high tumor expression of pCHK1 (S296) could serve as a useful biomarker [124]. It is worthwhile to mention that V158411 was never tested in humans. 

#### 3.2.4. Prexasertib

LY2606368 (prexasertib) is an ATP-competitive second-generation CHK1/2 dual inhibitor that mainly inhibits CHK1 and demonstrates single-agent activity in vitro and in vivo [125]. Biomarker analysis suggests TNBC cells with higher phosphorylation levels of DNA-PKcs or RPA32 and TNBC PDX tumors with higher mRNA expression levels of cyclin E1, cyclin D1, and MYC demonstrate higher sensitivity to prexasertib with regard to proliferation [126]. Clinical trials utilizing prexasertib in germline BRCA wildtype (BRCAwt) high-grade serous ovarian cancer (HGSOC) demonstrate activity and acceptable safety [127]. Given overlapping molecular features between subsets of sporadic TNBC with HGSOC, such as the high presence of TP53 mutations and genomic instability, prexasertib is anticipated to be similarly efficacious in TNBC models. Of note, no correlation was demonstrated between TP53 mutational status and sensitivity to LY2606368 [128]. LY2606368 is presently undergoing clinical development as a monotherapy treatment and in conjunction with both cytotoxic and targeted agents in solid tumors. A phase II pilot study by the National Cancer Institute (NCI) utilizing LY2606368 in BRCAwt patients who had at least one prior treatment (NCT02203513) resulted in one out of nine patients achieving PR (ORR of 11.1%) and four patients experiencing SD. The median PFS was 86 days (range 17 to 159 days). Grade 3/4 adverse effects, such as anemia, afebrile neutropenia, and thrombocytopenia, were reported, but they were manageable with supportive care measures [129]. Another phase II study utilizing LY2606368 in advanced solid tumors with *MYC* amplification, cyclin E1 amplification, *Rb* loss, or *FBXW7* mutations or tumors with HRR gene mutations is ongoing (NCT02873975) with the main objective of exploring the antiproliferative activity of prexasertib regarding these mutations. 

In TNBC cell lines, prexasertib treatment downregulates HRR proteins by promoting the ubiquitin proteasome-mediated degradation of BRCA1 and RAD51 [130]. Cells with HRR deficiency display increased DNA lesions and synthetic lethality upon treatment with PARPi [131], suggesting that prexasertib sensitizes TNBC cells to olaparib-induced DNA damage [130]. A phase I study of combination therapy of LY2606368 and olaparib for advanced solid tumors (NCT03057145) was completed in 2021, with results not yet reported. The evaluation of prexasertib plus samotolisib (LY3023414), which is a dual PI3K/mTOR inhibitor [132], showed the inhibition of primary tumor growth in preclinical models. A phase Ib clinical trial (NCT02124148) subsequently followed, exploring this combination. Common treatment-related adverse events were leukopenia/neutropenia (94.3%), thrombocytopenia (62.3%), and nausea (52.8%). Two patients achieved PR for an overall ORR of 15.4%, and ORRs were 25% for TNBC [133]. Currently, a phase II study (NCT04032080) evaluating the efficacy of samotolisib and prexasertib in patients with mTNBC is ongoing and was estimated to be completed by the end of 2023.

#### 3.2.5. UCN-01

UCN-01 (7-hydroxystaurosporine) was the pioneering CHK1 inhibitor to enter clinical trials, although it notably targets other serine–threonine protein kinases to include specific protein kinase C isoenzymes; Cdk2, 4, and 6; and PDK1 [134,135,136]. Preclinical studies demonstrated that UCN-01 potentiated the apoptosis-inducing effects of irinotecan in P53-mutated TNBC both in vitro and in vivo [137]. In a phase I study of UCN-01 (NCT00031681) in combination with irinotecan in patients with resistant solid tumor malignancies, two out of five women with p53-mutated TNBC achieved PR [138]. However, a phase II study of UCN-01 in combination with irinotecan in patients with mTNBC proved that this regimen had limited activity in TNBC. Only one among 25 mTNBC patients receiving irinotecan and UCN-01 achieved PR. Three patients experienced grade 3/4 adverse events, including neutropenia, anemia, nausea, and diarrhea, possibly due to limited bioavailability of UCN-01 to tissue, likely as a result of the high-affinity binding of this agent to the α1-acid glycoprotein (AAG) [138,139,140,141].

#### 3.2.6. MK-8776

Lastly, a recently described CHK1 inhibitor, MK-8776, was found to increase the radiosensitivity of human TNBC by inhibiting irradiation-induced autophagy in human TNBC cell lines. It remains to be seen if MK-8776 may demonstrate radiosensitization in clinical TNBC studies [142]. 

Clinical trials utilizing ATR and CHK1 inhibitors detailed above are presented in Table 1.

### 3.3. ATM-CHK2 Inhibitors in TNBC

As the role of CHK1 inhibitor-monotherapy and combination therapy with chemotherapy continues to evolve in the management of TNBC, ATM-CHK2 inhibitors have begun to emerge as potential agents of interest. ATM–CHK2 and ATR–CHK1 axes differentially respond to aberrant DNA structures. ATM is recruited to and primarily activated DSBs [144,145], whereas ATR is activated by recruitment to tracts of ssDNAs [146,147,148]. Germline ATM mutations confer an increased susceptibility to familial breast cancer and are associated with homologous recombination deficiency (HRD) in patients with BRCA1/2-wt breast cancer [149,150]. Accumulating preclinical studies suggest CHK2 inhibition could help overcome resistance to PARPi. The combination of ART-122, a small CHK2 inhibitor, and olaparib demonstrated a synergistic effect in BRCAwt ovarian and breast tumor cell lines. In the mouse xenograft model, the CHK2 inhibitor ART-122 sensitized BRCA1/2-wt tumor cells MDA-MB-231 to olaparib treatment. These data provide a strong rationale for the further development of ART-122 as a therapeutic agent currently limited to preclinical use [151]. 

Relating to the ATM-CHK2 pathway, AZD0156 is a potent and selective inhibitor of ATM and, in patient-derived TNBC xenograft models, was observed to enhance response to olaparib treatment by suppressing the repair of DNA damage caused by olaparib [152]. A phase I trial of AZD0156 as monotherapy or in combination with olaparib or FOLRIRI in patients with metastatic cancers was recently completed, and the results are pending (NCT02588105).

The ongoing clinical trials utilizing ATR, CHK1, and ATM-CHK2 inhibitors are presented in Table 2.

Also, Figure 2 depicts ATR and CHK1-ATM inhibitors and the drugs that these inhibitors are combined within TNBC (radiation therapy, chemotherapy, PARP inhibitors, and immunotherapy).

## 4. Discussion

ASCENT (NCT02574455) and DESTINY-Breast04 (NCT03734029) introduced sacituzumab govitecan and trastuzumab deruxtecan as novel therapeutic options in the landscape of TNBC management by improving PFS and OS compared to standard treatments [ASCENT: PFS 5.6 months with sacituzumab govitecan and 1.7 months with chemotherapy, OS 12.1 months and 6.7 months, respectively. DESTINY-Breast04: PFS 9.9 months in the trastuzumab deruxtecan group and 5.1 months in the physician’s choice group, OS 23.4 months and 16.8 months, respectively] and provided context for what defines a favorable outcome within the scope of the current options accessible to these patients [24,153]. Similarly, ATR and CHK1/2 inhibitors hold promise as potential cancer therapies for the management of TNBC by targeting the DDR pathway and disrupting cell cycle regulation in cancer cells. Preliminary results from preclinical and clinical studies demonstrate remarkable antitumor activity with a combination of ATR and CHK1/2 inhibitors with chemotherapy in TNBC models. Preclinical studies indicate that cells resistant to PARPi and deficient in BRCA show an elevated dependence on ATR signaling for stabilizing replication forks [154,155]. ATR as a target has the potential to overcome resistance to PARP inhibition. It is worthwhile mentioning that in the absence of a pathogenic DDR alteration, ATR inhibition is not a generally effective approach. While challenges remain, ongoing research and clinical trials continue to illuminate their therapeutic potential and refine their role for use in the treatment of TNBC.

While the potential utility of these agents in clinical practice was comprehensively reviewed herein, larger-scale clinical studies are needed to validate their benefit. Another logistic challenge lies in the selection process among the ATR inhibitors currently advancing in clinical development. For instance, M4344, similar to BAY1895344 and ceralasertib, is administered orally, while the first-generation ATR inhibitor berzosertib (M6620/VX-970) requires intravenous (IV) administration. IV delivery allows for more accurate drug dosing while oral administration facilitates a continuous treatment approach, aligning effectively with the concept of ‘gapped scheduling’. 

Selectivity and, thus, toxicity remain concerns, as these inhibitors will invariably affect non-neoplastic cells, leading to off-target effects. Information on adverse events related to ATR, CHK1, and CHK2 inhibitors is still accumulating. The inhibition of DDR inevitably induces on-target and off-target adverse events. Hematological and gastrointestinal adverse events are anticipated with all these compounds, but they are typically manageable through dose adjustments and supportive measures, as needed. The most frequent adverse effects related to ATR and CHK1/2 inhibitors are anemia, thrombocytopenia, and neutropenia, which are also common with chemotherapy administration. Hematologic toxicity was also observed with Olaparib [22] and sacituzumab govitecan [24]. However, targeting ATR might be a selective strategy for cancer cells but not normal cells, making ATR an attractive target compared to chemotherapy administration. Special attention should be given to rare but severe adverse events, such as pneumonitis or myelodysplastic syndrome/acute myeloid leukemia. Safety can pose a challenge when DDR targeting agents are combined with other molecules. Indeed, combinations with chemotherapy have proven to be considerably toxic in some cases, while combinations with anti-angiogenic agents, as well as with immune checkpoint inhibitors, were generally better tolerated [156]. Most of these novel treatments currently being evaluated have better response rates as compared with standard chemotherapy in heavily pretreated patients. They could be helpful in overcoming the issues posed by current treatments, such as the risk of autoimmune-related adverse events associated with immunotherapy [23]. However, a detailed comparison with the standard treatments cannot be made without the selection of patients with specific biomarkers. Subsequent translational research concentrating on factors predisposing to toxicity could enhance the precision of treatment personalization and aid in making informed therapeutic decisions.

Furthermore, resistance mechanisms will inevitably emerge, necessitating a thorough understanding of the emerging adaptive responses. EGFR overexpression and activation were documented as an innate resistance mechanism to prexasertib in TNBC and potentially other cancers. Using a panel of pre-clinical TNBC cell lines, the upfront evaluation of EGFR expression status considered in clinical trials was recommended [157]. It was further noted that although prexasertib monotherapy does not exhibit significant efficacy, its combination with an EGFR inhibitor demonstrates synergistic anti-proliferation effects. A separate preclinical study suggests EGFR inhibition may be used to overcome resistance to CHK1 inhibition [158]. 

Identifying predictive biomarkers that indicate response to these inhibitors is also expected to facilitate personalized treatment strategies but also better elucidate the underlying biology. Gene expression signature-related replication stress and neuroendocrine differentiation were associated with the prediction of response to M4344 therapy [78]. Preclinical and clinical studies supported ATM deficiency as a potential biomarker of sensitivity to ATR inhibition [98,159,160,161]. Apart from ATM deficiency serving as a predictive biomarker for ATR inhibitors, Williamson et al. previously identified defects in ARID1A as a synthetic lethal partner for ATR inhibition. Increased dependency on ATR-mediated cell cycle checkpoints and susceptibility to ATR inhibition were observed across various ARID1A-mutated models [162].

Recently, Pan et al. showed that Ring finger protein 126 (RNF126) promotes breast cancer metastasis. RNF126 functions as an E3 ubiquitin ligase, participating in a range of biological functions such as cell proliferation [163], DNA repair [164], and the regulation of the cell cycle [165]. RNF126-expressing breast cancer cells exhibit CDK2-mediated replication stress that makes them potential targets for ATR inhibitors. RNF126 also plays an important role in TNBC and its response to radiotherapy. Due to the limited availability of therapeutic options for TNBC, irradiation (IR) therapy remains a prevalent treatment choice for patients dealing with lymph node or brain metastasis [166]. E3 ubiquitin ligase RNF126 was reported to be important for IR-induced ATR-CHK1 pathway activation to enhance DDR. The depletion of RNF126 leads to increased genomic instability and IR sensitivity in both TNBC cells and mice, providing a promising target for improving the sensitivity of TNBC to radiotherapy [167]. 

In addition to the comprehensive landscape of inhibitors reviewed herein, novel approaches to target ATM/ATR/CHK1 continue to be explored and developed, such as peptide drug conjugates that involve coupling a peptide with a cytotoxic drug. This approach enables the drug to specifically target cells within the acidic pH environment of the tumor. In a preclinical in vivo study using human xenografts with HRD, ATRi ceralasertib displayed a synergistic effect in tumor growth suppression with CBX-12, a peptide drug conjugate that combines a pH-sensitive peptide with the potent topoisomerase I inhibitor exatecan [168]. Combinatorial approaches utilizing different mechanisms of action may more effectively overcome resistance mechanisms. Another promising therapeutic option for TNBC is immunotherapy, which has already shown encouraging efficacy [169]. Combining immune cell therapy with DDR inhibitors, such as ATR or CHK1/2 inhibitors, for the treatment of TNBC is also an area of interest.

## 5. Conclusions

Targeted cancer treatments mark a significant advancement in personalized medicine as they function via the inhibition of cancer-specific alterations. ATR and CHK1/2 inhibitors represent promising therapeutic options for TNBC, leveraging the vulnerabilities associated with genomic instability and dysregulated cell cycle checkpoints. While challenges persist, the synergistic potential of these inhibitors with existing treatments offers hope for improved outcomes in TNBC patients. Ongoing phase I and II clinical trials and further mechanistic elucidation through the evolution of phase III clinical trials will guide their integration into the evolving landscape of TNBC therapy.

## Figures and Tables

**Figure 1 cancers-16-01139-f001:**
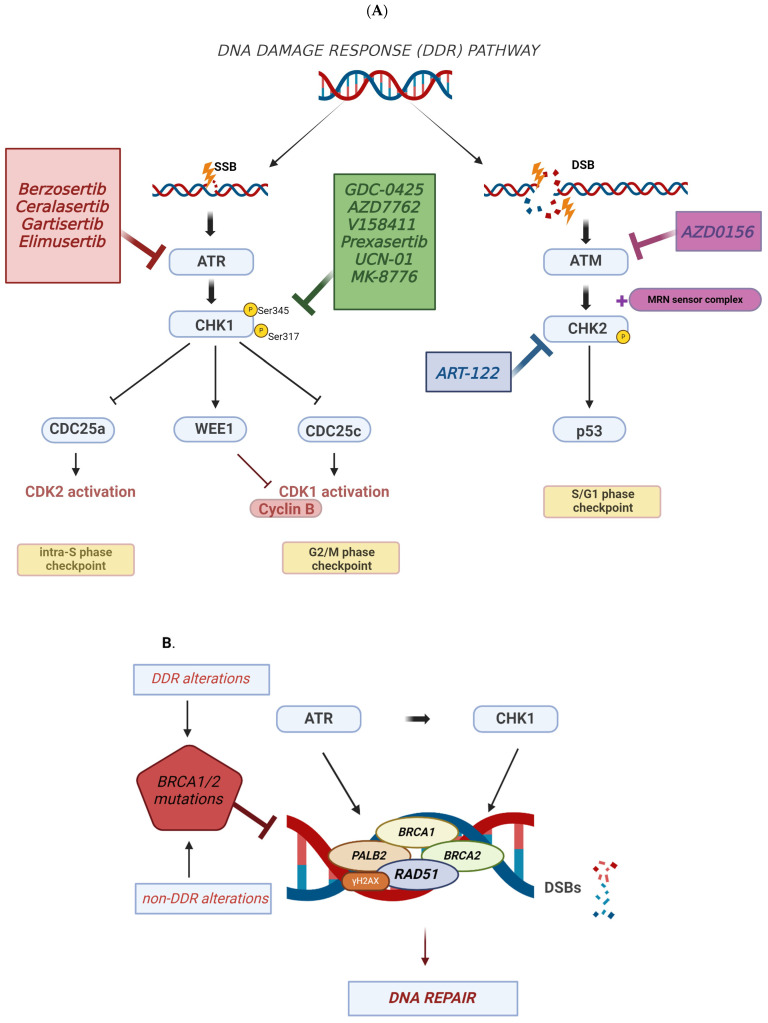
Targeting DDR in TNBC. (**A**) ATR-CHK1-WEE1 and ATM-CHK2 pathways. Square boxes show potential therapeutic inhibitors and their site of action in the DDR cascade. The inhibitors that are underlined are still under development in preclinical models. DNA damage activates DDR pathway initiated by ATR and ATM. Once activated, CHK1 obstructs S phase and G2/M phase progression by suppressing CDC25a and CDC25c, respectively. Activated ATR-CHK1-WEE1 induces G2-M cell cycle arrest by inhibiting the cyclin B-CDK1 complex to prevent cells with DNA damage from progressing through the cell cycle. (**B**) HRR pathway. ATR promotes HRR by phosphorylating BRCA1 and CHK1 and activation of RAD51 and BRCA2. RAD51 filament is responsible for DNA repair. HRR is ineffective in presence of BRCA1/2. Non-DDR alterations also lead to HRR deficiency (created with Biorender.com, accessed on 29 February 2024).

**Figure 2 cancers-16-01139-f002:**
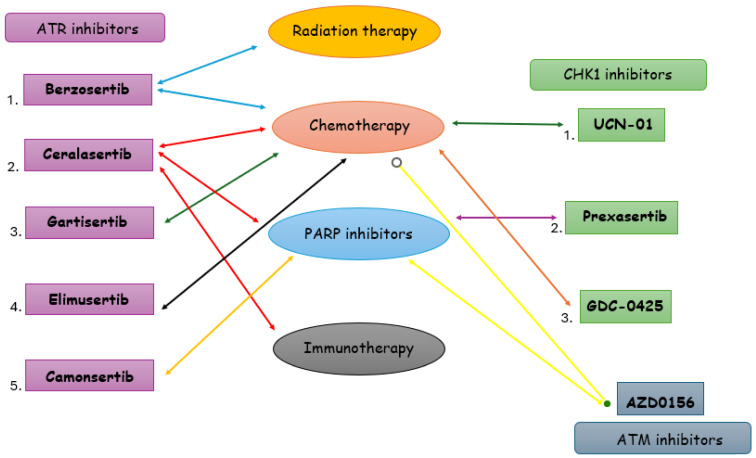
ATR and CHK1-ATM inhibitors and the drugs that these inhibitors are combined with in TNBC: radiation therapy, chemotherapy, PARP inhibitors, and immunotherapy.

**Table 1 cancers-16-01139-t001:** Completed clinical trials about ATR and CHK1 inhibitors.

DDRi	Agent	NCT ID Number	Status	Disease	Phase	Patient Number	Treatment	Safety	Result
ATRi	VX-970/Berzosertib	NCT02157792	Completed	BC	I	4 (cohort 1)	berzosertib iv (140 mg/m^2^; D2, D9), +cisplatin (75 mg/m^2^; D1) q21 days	Most common grade ≥ 3 TRAEs: neutropenia (20.0%) and anemia (16.7%).	1 TNBC patient achieved PR but progressed after 17 months [82]
			Completed	mTNBC	I	47 (cohort 2)	berzosertib iv (140 mg/m^2^; D2, D9), +cisplatin (75 mg/m^2^; D1) q21 days	Grade ≥ 3 TEAEs: neutropenia (38.3%), anemia (25.5%), thrombocytopenia (12.8%), and vomiting (12.8%).	ORR: 11/47 (23.4%) patients [84]
			Completed	Refractory Solid Tumors	I	2 (cohort 3)	berzosertib iv 90 mg/m^2^ (D2, D9) + carboplatin AUC 5 (D1) q21 days	Grade ≥ 3 AEs: neutropenia (21.7%), anemia (4.3%), and thrombocytopenia (4.3%).	1 mBC patient experienced dose-limiting febrile neutropenia (berzosertib 90 mg/m^2^ once weekly + carboplatin) [85]
	AZD6738/Ceralasertib	NCT02264678	Recruiting	Advanced Solid Malignancies	I/Ib	Still recruiting	ceralasertib (in escalating doses 60 mg OD to 240 mg BD D5–D14) + carboplatin, Olaparib, or durvalumab	Frequent TRAEs: thrombocytopenia, anemia, neutropenia, fatigue, decreased appetite, nausea, vomiting, constipation, diarrhea, cough.	FIRST RESULTS: 2 BRCA mutant TNBC patients achieved PRs [143]
		VIOLETTE (NCT03330847)	Active, Not Recruiting	mTNBC	II	226	olaparib (300 mg BID q28 days) + ceralasertib (160 mg D1–D7) q28 days vs. olaparib (300 mg BID q28 days) monotherapy	Most common TRAEs: nausea and anemia (47% of patients).	No significant difference in ORR or PFS with the combination [89]
	VX-803/M4344/Gartisertib	NCT02278250	Completed	Advanced Solid Tumors	I	97	Gartisertib (250 mg qd) monotherapy or in combination with carboplatin	Significant liver toxicity [95].	n/a
	RP-3500/Camonsertib	NCT04497116	Active, Not Recruiting	Advanced Solid Tumors harboring DDR mutations	I	120	Camonsertib orally (160 mg D1–D3 qw) monotherapy	Anemia was the most common drug-related toxicity (32% grade 3).	Upon camonsertib treatment in 1 patient with BRCA1-mutated TNBC, all BRCA1 mutations declined in blood but then rebounded before progression of non-target lesions at 29 weeks of treatment. Overall, anemia was the most common drug-related toxicity (32% Grade 3) [79]
CHK1i	GDC-0425	NCT01359696	Completed	Refractory Solid Tumors	I	5	GDC-0425 orally (60 mg). +gemcitabine iv (1000 mg/m^2^)	DLTs: thrombocytopenia (*n* = 5), neutropenia (*n* = 4), dyspnea, nausea, pyrexia, syncope, and increased. alanine aminotransferase (*n* = 1 each). Common TRAEs: nausea (48%); anemia, neutropenia, vomiting (45% each); fatigue (43%); pyrexia (40%); and thrombocytopenia (35%).	2 PRs in TP53 mutated TNBC [112]
	LY2606368/Prexasertib	NCT02203513	Terminated	BRCAwt patients	II	9	Prexasertib iv 105 mg/m^2^ q2w monotherapy	Grade 3/4 TRAEs: afebrile neutropenia (*n* = 8; 88.9%), anemia (*n* = 3; 33.3%), and thrombocytopenia (*n* = 1; 11.1%).	1 PR and 4 SD [129]
		NCT02873975	Completed	Advanced Solid Tumors with specific mutations	II	n/a	n/a	n/a	n/a
		NCT03057145	Completed	Advanced Solid Tumors	I	n/a	Prexasertib + Olaparib	n/a	n/a
		NCT02124148	Completed	Advanced and/or Metastatic Solid Tumors	Ib	53	Prexasertib iv (105 mg/m^2^ q14 days) + Samotolisib orally (200 mg twice daily)	Common TRAEs: leukopenia/neutropenia (94.3%), thrombocytopenia (62.3%), and nausea (52.8%).	ORR for TNBC: 25% [133]
	UCN-01	NCT00031681	Completed	Resistant Solid Tumor Malignancies	I	5	UCN-01 iv (70 mg/m^2^ D2, 35 mg/m^2^ D23) +irinotecan iv (125 mg/m^2^ D1, D8, D15, D22)	DLTs: grade 3 diarrhea/dehydration and dyspnea.	2 out of 5 PRs in p53 mutated TNBC [138]
		n/a	Completed	mTNBC	II	25	UCN-01 + irinotecan	n/a	1 PR [138,139,140,141]

AUC = Area under curve, BC = breast cancer, BID = twice a day, DLT = dose-limiting toxicity, mTNBC = metastatic TNBC, ORR = overall response rate, PFS = Progression-free survival, PR = partial response, SD = stable disease, TNBC = triple-negative breast cancer.

**Table 2 cancers-16-01139-t002:** Ongoing clinical trials on ATR, CHK1, and ATM-CHK2 inhibitors.

DDRi	Agent	NCT ID Number	Disease	Phase	Treatment
ATRi	VX-970/Berzosertib	NCT04052555	Chemotherapy resistant TNBC/HR+, HER2-BC	Ib	Berzosertib + Radiation Therapy
	AZD6738/Ceralasertib	NCT04090567	Germline BRCA-mutated BC advanced or metastatic	II	Olaparib + Cediranib or AZD6738
		PHOENIX (NCT03740893)	Neoadjuvant chemotherapy-resistant residual TNBC	IIa	short exposure to DDRi and/or anti-PD-L1 immunotherapy preoperatively
		NCT03801369	mTNBC	II	Olaparib + durvalumab, selumetinib, or capivasertib or ceralasertib alone
	BAY-1895344/Elimusertib	NCT04491942	Solid Tumors	I	Elimusertib + chemotherapy
	RP-3500/Camonsertib	NCT04972110	Advanced Solid Tumors	I/II	Camonsertib + Niraparib/Olaparib
CHKi	LY2606368/Prexasertib	NCT04032080	mTNBC	II	Prexasertib + Samotolisib
ATMi	AZD0156	NCT02588105	Metastatic disease	I	AZD0156 as monotherapy or +olaparib or FOLRIRI

BC = breast cancer, mTNBC = metastatic TNBC, TNBC = triple-negative breast cancer.

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
