# Peer review of "Triple-Negative Breast Cancer and Emerging Therapeutic Strategies: ATR and CHK1/2 as Promising Targets"

_cancers, 2024, doi:10.3390/cancers16061139_

Round 1

Reviewer 1 Report

Comments and Suggestions for Authors

The manuscript by Sofianidi et al. presents a comprehensive review of the potential of ATR and CHK1/2 inhibitors as novel therapeutic approaches for triple-negative breast cancer (TNBC). The authors highlight the aggressiveness of TNBC and the scarcity of treatment options, summarizing the role of the DNA Damage Response (DDR) pathway in TNBC. They delve into the potential of targeting dysregulated DDR with ATR and CHK1/2 inhibitors in TNBC, detailing the mechanisms of action of these inhibitors, as well as their preclinical and clinical evaluations, and the challenges faced in their development and administration.

The manuscript's strength lies in the effective discussion of the DDR pathway's role in TNBC and the promising results of ATR and CHK1/2 inhibitors in preclinical and clinical studies. However, the manuscript could be enhanced by: 1) providing a more detailed discussion of the specific challenges encountered in key clinical trials; 2) offering a deeper analysis of the comparative effectiveness of these inhibitors against standard treatments and their integration into current therapeutic regimens for added insight; and 3) addressing the potential side effects and toxicity profiles of these inhibitors to improve the manuscript's comprehensiveness.

A minor point regarding the title, “ATR and CHK1/2 Inhibitors as Promising Targets,” feels awkward. A more fluent alternative could be “ATR and CHK1/2 as Promising Targets” or “Inhibitors as Promising Therapeutics.”

Author Response

We would like to thank the Reviewer for his/her thoughtful evaluation of our manuscript and for the most welcome comments/suggestions. Accordingly, we have now revised our manuscript to reflect these comments.

In the revised Text all changes/additions/modifications made in response to the Reviewers’ points ((including newly added references in text and reference list) are highlighted in red.

Please find below a point-by-point RESPONSE to the issues raised by the Reviewer:

The manuscript by Sofianidi et al. presents a comprehensive review of the potential of ATR and CHK1/2 inhibitors as novel therapeutic approaches for triple-negative breast cancer (TNBC). The authors highlight the aggressiveness of TNBC and the scarcity of treatment options, summarizing the role of the DNA Damage Response (DDR) pathway in TNBC. They delve into the potential of targeting dysregulated DDR with ATR and CHK1/2 inhibitors in TNBC, detailing the mechanisms of action of these inhibitors, as well as their preclinical and clinical evaluations, and the challenges faced in their development and administration.

The manuscript's strength lies in the effective discussion of the DDR pathway's role in TNBC and the promising results of ATR and CHK1/2 inhibitors in preclinical and clinical studies;

RESPONSE: We thank the Reviewer for his/her positive evaluation and kind words regarding our work.

However, the manuscript could be enhanced by: 1) providing a more detailed discussion of the specific challenges encountered in key clinical trials

RESPONSE: We agree with the Reviewer and according to his/her suggestion, we added a column in Table 1 under the name “Safety”, where we present the challenges and the adverse events encountered in the clinical trials. Also, we have added some information regarding safety in the text.

2) offering a deeper analysis of the comparative effectiveness of these inhibitors against standard treatments and their integration into current therapeutic regimens for added insight

RESPONSE: We agree with the Reviewer, and we added a brief reference regarding the comparison of these novel drugs with standard treatment in the Discussion part.

3) addressing the potential side effects and toxicity profiles of these inhibitors to improve the manuscript's comprehensiveness.

RESPONSE: We tried to address this comment alongside with comment 1 regarding safety and toxicity.

A minor point regarding the title, “ATR and CHK1/2 Inhibitors as Promising Targets,” feels awkward. A more fluent alternative could be “ATR and CHK1/2 as Promising Targets” or “Inhibitors as Promising Therapeutics.”

RESPONSE: We agree with this comment and we changed the title to “Triple Negative Breast Cancer and Emerging Therapeutic Strategies: ATR and CHK1/2 as Promising Targets”.

Reviewer 2 Report

Comments and Suggestions for Authors

In the present manuscript author try to discuss novel strategies to control TNBC. So far overall manuscript look good and well structure. Full manuscript written well but few major concern need to address before further process. Such as author need to compare with current therapy vs ATR and CHK1/2 inhibitor. More need to understand the safety of this inhibitor as well as pharmacokinetic parameter. Also need to understand which kind of formulation is more efficacious in TNBC like small or large molecule with justification. Lastly author need to put more diagrammatic  representation for better reader view. 

Author Response

We would like to thank the Reviewer for his/her thoughtful evaluation of our manuscript and for the most welcome comments/suggestions. Accordingly, we have now revised our manuscript to reflect these comments.

In the revised Text all changes/additions/modifications made in response to the Reviewers’ points ((including newly added references in text and reference list) are highlighted in red.

Please find below a point-by-point RESPONSE to the issues raised by the Reviewer:

In the present manuscript author try to discuss novel strategies to control TNBC. So far overall manuscript look good and well structure.

RESPONSE: We thank the Reviewer for his/her positive evaluation and kind words regarding our work.

Full manuscript written well but few major concern need to address before further process. Such as author need to compare with current therapy vs ATR and CHK1/2 inhibitor.

RESPONSE: We agree with the Reviewer, and we added a brief reference regarding the comparison of these novel drugs with standard treatment in the Discussion part.

More need to understand the safety of this inhibitor as well as pharmacokinetic parameter. Also need to understand which kind of formulation is more efficacious in TNBC like small or large molecule with justification.

RESPONSE: We agree with the Reviewer and according to his/her suggestion, we added a column in Table 1 under the name “Safety”, where we present the challenges and the adverse events encountered in the clinical trials and we have added some information regarding safety in the text.  Also, we added the recommended phase II dose and schedule under the Treatment column.

 Lastly author need to put more diagrammatic  representation for better reader view. 

RESPONSE: We agree with the Reviewer and in order to address his/her comment, we added a second figure regarding ATR and CHK1-ATM inhibitors and the drugs that these inhibitors are combined with (radiation therapy, chemotherapy, PARP inhibitors and immunotherapy). Also, we modified Figure 1.

Round 2

Reviewer 2 Report

Comments and Suggestions for Authors

It is now accept for publication process.